# Measures to Cope with the Impact of Climate Change and Drought in the Island Region: A Study of the Water Literacy Awareness, Attitude, and Behavior of the Taiwanese Public

**Jo-Hung Yu [1], Hsiao-Hsien Lin [2], Yu-Chih Lo [2], Kuan-Chieh Tseng [3] and Chin-Hsien Hsu [2],***

[1] Department of Marine Leisure Management, National Kaohsiung University of Science and Technology, Kaohsiung 811034, Taiwan; henry@nkust.edu.tw

[2] Department of Leisure Industry Management, National Chin-Yi University of Technology, Taichung 41170, Taiwan; chrishome12001@yahoo.com.tw (H.-H.L.); loyuchih@ncut.edu.tw (Y.-C.L.)

[3] MA Program in Social Enterprise and Cultural Innovation Studies, College of Humanities & Social Sciences, Providence University, Taichung 433719, Taiwan; jackt72@pu.edu.tw

* Correspondence: hsu6292000@yahoo.com.tw

**Abstract:** This study assessed people's water literacy awareness, attitudes, and behaviors to identify strategies for coping with drought and water scarcity. The data from 653 questionnaires were analyzed by statistical validation and using IBM SPSS 22 and IBM AMOS 26.0. The views of students, housewives, swimming pool owners, schoolteachers, and experts were collected and finally examined by multivariate validation analysis. People have a high level of water literacy and developed sufficient water-saving habits (4.60). Although most people believe that tap water is of good quality, it is difficult to deliver and expensive, and cannot be consumed directly. Even though people are aware of the water shortage crisis, willing to carry water bottles instead of using plastic bottled water, choosing to buy environmentally friendly cleaning products (4.08), performing water conservation behaviors on the go, taking showers within 6–15 min, and taking the initiative to notify the relevant authorities to repair water facilities, the frequency of using bottled water is still high due to work and living habits, consumption ability, and mobility constraints (34.6), and they are less willing to buy products with the "water proficiency label" (4.08) and participate in stream-cleaning activities (3.57). The willingness to participate in water purification activities is low. The public also feels that the government is responsible for solving the current water shortage crisis (3.71). There are significant differences in the perceptions, attitudes, and behaviors of water literacy among people of different genders, ages, and regions, depending on their work and consumption abilities, quality of life, and convenience ($p < 0.05$). Increasing water responsibility can enhance environmental management actions, consumer economic actions, and civic actions, while enhancing water perceptions and crisis awareness can further strengthen civic behaviors.

**Keywords:** climate change; drought; water sustainability; water literacy; governance effectiveness

## 1. Introduction

Under the influence of global climate change, abnormal temperatures lead to the gradual disappearance of the Antarctic and Arctic ice caps and glaciers [1], rising sea levels, and increasing water temperatures, resulting in abnormal climate, frequent heavy rainfall and droughts [2], irregular rainfall, deranged freshwater regulation, and the gradual depletion of water resources [3], leading to a crisis of human survival.

Deforestation leads to increased surface albedo, reduced tropical evapotranspiration, lower water content, and decreased global average rainfall [4]. This, together with fuel combustion and emissions for energy production [5], has led to the phenomenon of global climate change. This phenomenon has a great impact on the hydrological system of the world, including Taiwan [6,7]. Climate change will alter the regional climate, temperature, humidity, and atmospheric pressure, which in turn will affect rainfall and lead to

drought [4,8]. The effects will lead to land degradation, desertification, gradual loss or death of forests and vegetation [8,9], increased wildfires and pests, and shortage of fresh water and food, all of which threaten human survival [10]. Taiwan, located in the Asian region and surrounded by sea, is a typical island climate area [6]. While there are rivers and streams in the east and west of Taiwan, and 21 reservoirs have been built to store 411,896,200 tons of fresh water [7], which can provide for the population of 23,560,000 [8] and various industries in Taiwan, the average daily water consumption per person in Taiwan is 276 L [9], which translates to a total consumption of 237,343,400 tons of fresh water per year for the entire population in Taiwan. If we do not take into account the annual rainfall, the annual water consumption of the people in Taiwan has already depleted by about 57.62% of the total storage capacity. Even with the distribution of streams and rivers and the construction of reservoirs in various regions, the rainfall is still unable to overcome the problem of climate change, and the rainfall has dropped dramatically, bringing the major reservoirs to the brink of depletion [7,11,12]. The problem of depleted reservoirs and impending water shortage has received great attention from the government and the people of Taiwan.

Since 2012, there has been a trend of environmental deterioration due to the increase in environmental pollution concentration and other activities and processes. Coupled with extreme weather events, climate change, and the failure of regulatory measures, the water crisis has escalated from a purely environmental issue to a social issue [13]. Forty percent of the world's population is still affected by the water crisis. The United Nations has, therefore, adopted 17 indicators for sustainable development to ensure that all people have access to clean water and health resources [14]. The goal is to protect and restore water-related ecosystems by 2020, and to fully implement integrated, transboundary water management measures by 2030. This will ensure sustainable freshwater supply and recycling, reduce waste and wastewater discharge, and propose ways to improve water quality [15]. From 2001 to 2010, Taiwan experienced water supply problems due to insufficient rainfall or heavy rainfall that produced turbidity in water quality [16]. Coupled with the siltation of reservoirs, over-pumping of groundwater, and industrial pollution, multiple water resource problems remain to be solved [17]. Recently, scholars investigated the blue water footprints of various counties and cities in Taiwan to identify their water consumption patterns and strategies [18]. Their suggestions included the conservation and management of reservoir catchment areas, upgrading and improving facilities, strengthening the allocation of regional water resources, improving the efficiency of agricultural, industrial, and domestic water use, water conservation, diversified water resources development [19], revising regulations [20], improving household wastewater-recycling technology [21], and improving water management measures in the east [22]. However, the people also faced a water shortage crisis during 2009–2021 February [11,12]. Of course, a large part of this is due to the abnormalities of climate regulation and the sudden decrease in rainfall [7,11]. However, with the current technology, humans are not yet able to intervene or dominate the climate factor. In addition, due to the strong tourism development in Taiwan, people of different nationalities are staying or living here for a short period, but the population structure is still dominated by local residents [8]. Long-term residents are one of the largest users of water resources [7]. Therefore, in this study, we aimed to understand the freshwater consumption factors by looking at the water demand of individual water users or organizations in Taiwan, so as to explore the freshwater use habits and behaviors of water users. In the end, the study analyzed the public's awareness, attitude, and behavior toward freshwater use, and identified improvement measures and suggestions for water users. The results of the study will help the government and the public to overcome the crisis of freshwater shortage in the present and future drought crisis, which is the purpose of this study.

## 1.1. Water and the Activities and Health of the People of Taiwan

Water is one of the resources that human beings depend on for their livelihood. However, heavy rains and droughts caused by climate change [23] can lead to inconvenience, infectious diseases [24], food shortage, hunger [25], and even death. Long-term observations show that excessive or lack of water resources can cause social instability and affect human physical and mental health [26].

As an island country surrounded by sea, Taiwan is prone to storm attacks and the short rivers and waterways do not retain water resources easily, making it susceptible to floods and droughts [16]. The phenomenon of flooding or drought can lead to different diseases, and heavy rainfall can cause most infectious diseases (except hepatitis A and enterovirus infections). After heavy rainfall or drought, pathogenic mosquitoes can spawn [27], which can cause physical and psychological damage to humans [28]. In particular, the drought crisis has emerged recently due to the scarcity of rainfall [11,12], which will be a major crisis for the people of Taiwan, who consume a large amount of fresh water.

However, the water that bears the boat is the same that swallows it up, and this is also true for the people of Taiwan who are short of water. In Taiwan, due to the topographic constraints, freshwater resources are not easily saved, and the people use a huge amount of water every year, consuming almost more than half of the freshwater reserves [9]. Therefore, if it is not possible to increase the source of water and gain access to a large amount of rainfall, it would be helpful to start reducing the flow of water and find ways to conserve water by emphasizing people as the root cause to overcome the crisis.

Implementing energy and carbon reduction and improving people's knowledge and behavior about water are important issues today [29]. With the rapid increase in population, industrial development, and agricultural irrigation needs, as well as changes in the water environment caused by climate change, household water supply is becoming scarce [13,23]. There are many mountainous areas in Taiwan, and the rivers are sloping and fast flowing, so it is not easy to retain freshwater resources. The uneven temporal and spatial distribution of precipitation makes a big difference in water resources during the dry and abundant water periods [30]. Until there is rainfall, it is imperative to conserve water and reduce internal consumption.

Developing water literacy fosters environmental citizens who reflect, care, think, evaluate, and act critically [31], and who are able to make thoughtful decisions and act responsibly when faced with a water crisis [32,33]. This can promote energy and water conservation and achieve the goal of sustainable water resources management and human coexistence [34].

Hence, the researcher believes that until sufficient rainfall is available in Taiwan, we should take this opportunity to study the water use patterns, behaviors, and attitudes of the Taiwanese people to find ways to reduce the large amount of freshwater consumption in order to cope with the drought crisis and to construct water management measures with the goal of energy conservation and sustainability, which will help to formulate countermeasures when facing the problem of water scarcity in the future.

## 1.2. Environmental Citizenship Behavior

The purpose of environmental education is to develop responsible citizenship, where education is the process and deep-rooted literacy is the outcome [35]. Environmental literacy is expected to shape an individual's ability to discern and act on the current state of wellness of environmental systems [36], and to develop individuals with personal emotional interactions, attitudes, beliefs, identities, knowledge, worldviews, values, skills, as well as the basic concepts and awareness to take action [37].

Environmental citizenship behavior is a model of voluntary behavior [38] that contributes to the sustainability of organizations, societies, and nations through individual efforts [39]. In the process of action, individuals are willing to cooperate with others if the cooperation exceeds the expected results [40]. There is no need for rewards or other incentives, but only a desire to build on their own well-being and to create more value [41].

Therefore, it is believed that environmental citizenship behavior is a behavior pattern in which people, individually or in cooperation with others, spontaneously carry out the cleaning of the community or the natural environment to achieve personal well-being and sustainable development of society and the nation, without incentives. In this way, individuals develop a willingness to work for the sustainability of the natural environment, implement beach cleaning, and control their own water use habits [23] in order to achieve the goal of water conservation and resource sustainability.

*1.3. Water Literacy*

Taiwan is surrounded by sea. Freshwater resources from natural precipitation are one of the major sources of water for domestic use. However, the collection of freshwater resources is not easy due to the steep terrain [16]. In addition, Taiwan is facing a period of drought and the freshwater resources stored in reservoirs will be depleted, which will seriously affect people's livelihoods and industrial development [11,12]. Therefore, water conservation is the best way to overcome Taiwan's water shortage crisis. The key to achieving the goal of water conservation and effective policy implementation is the users of the resource [41]. Getting people to adopt proper water use can help achieve water conservation goals [22], and fostering water literacy is the smartest way to do so.

Water literacy is the ability to appreciate the value of water, to understand the operational characteristics of water supply systems, and to use water resources in the wisest way possible in the face of water crises [22]. For the public, water literacy is the ability to understand the importance of water conservation actions and good management, to recognize the need to take specific actions to protect water resources and establish their value, and to communicate this information effectively and clearly, which is known as national water literacy [37].

Water literacy can be subdivided into cognition, attitudes, and behaviors. Water cog-nition refers to building awareness and developing an understanding of water issues and challenges to act, express opinions, and collaborate on water issues locally and world-wide [42–44]. Evidently, having the right water cognition should help the Taiwanese people form a correct consensus on water conservation and overcome the water shortage crisis. However, since the people of Taiwan consume a large amount of water for domestic use, it will be important to explore the level of water literacy awareness among the people. Proper water cognition can be examined in terms of basic water knowledge and local water resources [22].

Water attitudes are deeply rooted in the issues of sustainability, maintenance, and importance of water, and enhance individuals' attitudes and awareness of the importance of water conservation [45,46]. Water conservation requires a common belief and consensus to help people embrace water conservation measures and improve the effectiveness of policy decisions. The right attitude toward water conservation can be explored in terms of water values, responsibilities, and water ethics [22,47].

Water behaviors refer to the willingness of individuals to spontaneously adopt public behaviors or actions that are beneficial to water resources [48], such as water conservation devices or water use frequency, after recognizing the water resources. Water conservation measures are not just slogans, they require practical action. Having the right water conservation behavior is the only way to achieve the goal of water conservation. Effective water conservation behavior can be explored in terms of appreciation of water resources, responsibility, and water ethics [22].

The establishment of good water literacy helps people to understand the importance of water conservation actions and good management, to recognize and take certain actions to protect water resources and establish their value, and to convey relevant messages effectively and clearly [37], which helps to promote the correct use of water resources [22].

Therefore, this study believes that addressing water literacy can help us understand the awareness, attitudes, and behaviors of major freshwater users regarding the conservation of water resources, and obtain information about people's daily water use habits so that we can

further estimate the loss of water resources, provide the government with the opportunity to make energy-saving and water-saving decisions, and raise people's awareness of the problem of overuse or consumption of water. Understanding the impact of water literacy awareness, attitudes, and behaviors will help enhance the correct water use behaviors of people.

## 2. Methods and Instruments

### 2.1. Study Framework and Hypotheses

The study took reference to literature related to water literacy and adopted the concept of blue water footprint [1–48] to compile the questionnaire tools of water awareness [22,45], attitude [45–47], and behavior [22,48] to explore the water awareness, attitude, and behavior of Taiwanese people toward water resources. Currently, about 23.5 million people in Taiwan are facing the water crisis together [49]. Scholars believe that obtaining a valid sample size of 300 is sufficient for factor analysis [50], while obtaining more than 600 questionnaires with a sampling error of 5% and a confidence level of 95% is sufficient to represent a population of more than 100,000 [51,52]. First, three experts in environmental engineering management, soil and water conservation, and environmental education were invited to review the first draft of the questionnaire, and then 50 questionnaires were distributed and validated using IBM SPSS 22 and IBM AMOS 26.0 packages to obtain the final official questionnaire. A total of 653 questionnaires were collected from all regions of Taiwan between January and February 2021 using the convenience sampling method on the online questionnaire platform, and the sample data were analyzed using basic statistical validation and ANOVA. The results of the questionnaires were then used to seek the opinions of students, housewives, swimming pool operators, school teachers, as well as experts and scholars, and the information was compiled, organized, and analyzed in a rigorous sequence to construct the final content [53], which was analyzed and explored by multivariate validation [54,55], As shown in Figure 1. Based on the above, the research framework is as follows:

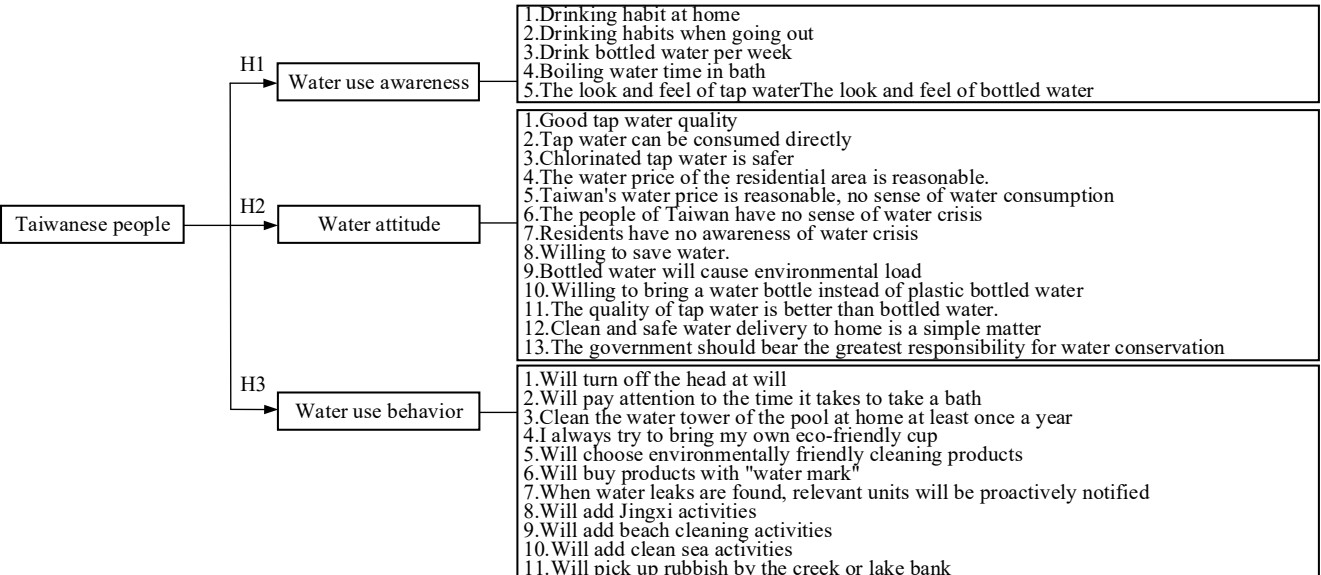

**Figure 1.** The research structure.

Based on the above description, there are 3 research hypotheses:

**Hypothesis 1 (H1).** *There is no change in the water cognition of the public in the crisis of water shortage.*

Water cognition is the basis for integrating perspectives and building cooperation when people are confronted with water issues [43–45]. Adequate water cognition should help the people of Taiwan to reach a consensus on water conservation in the face of the current water shortage problem. However, statistically speaking, people use a lot of water themselves. Therefore, there may be no difference in the perception of the impact of water scarcity in the current water crisis. Hence, we hypothesize that the water cognition of the public has not changed, even under the crisis of water shortage.

**Hypothesis 2 (H2).** *There is no change in the water attitude of the public in the crisis of water shortage.*

Appropriate water attitudes can help the people of Taiwan to develop awareness and attitudes toward water conservation in the face of water scarcity [45,46]. However, since water consumption remains high under normal conditions, it is unlikely that demand for water will change, even in the face of the current water crisis. Hence, we hypothesize that the water attitude of the public has not changed, even under the crisis of water shortage.

**Hypothesis 3 (H3).** *There is no change in the water behavior of the public in the crisis of water shortage.*

Proper water behavior can reduce the waste of local water resources [48] and effectively reduce water consumption. However, statistically, Taiwanese people are accustomed to consuming large amounts of water [7]. Hence, we hypothesize that the water behavior of the public has not changed, even under the crisis of water shortage.

*2.2. Study Procedure and Instruments*

The study focused on the water literacy cognition, attitudes, and behaviors of Taiwanese people. We read literature related to water literacy and environmental education [1–48], and then compiled questionnaire tools with reference to findings on water use awareness [22,45], attitudes [45–47], and behaviors [22,48]. The preliminary draft of the questionnaire was reviewed by three experts in the fields of environmental engineering management, soil and water conservation, and environmental education. Fifty questionnaires were distributed and tested for reliability using IBM SPSS 22 and IBM AMOS 26.0 packages. Cronbach's $\alpha$ of 0.7 or higher indicates that the question has high reliability [56] and sufficiently representative for subsequent analysis. However, due to the environmental safety of the COVID-19 epidemic and the extensive area of Taiwan, it was not easy to collect samples, so the online questionnaire platform was used to distribute the questionnaires by a convenience sampling method to obtain a representative number of valid formal questionnaires [57], and a total of 653 questionnaires were analyzed by statistical testing, ANOVA testing, and Pearson product-moment correlation coefficient.

There were 6 questions in the water use awareness questionnaire, and statistical analysis showed that the Kasier-Meyer-Oeasure (KMO) was 0.437, while Bartlett's approximate $\chi^2$ value was 38.019 and df was 15, with a significance of $p < 0.001$, which was suitable for factor analysis. The explained variance of the scale was 13.05%, and the total explained variance was 13.05%. After factor analysis, all of them were retained. The questionnaire was named as water awareness questionnaire with 6 questions and the alpha coefficient was 0.700, and the alpha coefficient of the whole questionnaire was 0.701. Based on the results of the above analysis, the reliability of this questionnaire was good.

There were 13 questions in the water attitudes questionnaire, and the results of the statistical analysis showed that the KMO was 0.642, and Bartlett's approximate $\chi^2$ value was 1230.509 with a df of 78 and a significance of $p < 0.001$, which was suitable for factor analysis. The explained variance of the scale was 12.13%, 9.22%, and 8.76%, and the total explained variance was 73.97%. After factor analysis, all of them were retained. The alpha coefficients of the four scales were 0.700, 0.701, and 0.705, respectively, and the alpha

coefficient of the total scale was 0.705. Based on the above analysis results, it was concluded that this questionnaire had good reliability.

There were 11 questions in the water behaviors questionnaire, and the results of statistical analysis showed that the KMO was 0.869, while Bartlett's approximate χ2 value was 3910.033 and df was 55, with a significance of $p < 0.001$, which was suitable for factor analysis. The explained variance of the scale was 28.23%, 18.12%, and 15.28%, and the total explained variance was 61.64%. After factor analysis, all of them were retained. The alpha coefficients of the three scales were 0.700, 0.701, and 0.705, and the alpha coefficient of the total questionnaire was 0.920. Based on the results of the above analysis, it was concluded that this questionnaire had good reliability. As shown in Table 1.

**Table 1.** Analysis of questionnaire tools.

| Construct | Dimension | |
|---|---|---|
| Basic Variables | gender (male: female), age (20 down: 21–30:31–40:41–50:51–60:61 up), Place of Residence (Northern:Central:Southern:Eastern:Outlying Islands) | |
| **Construct** | **Dimension** | **Cronbach's α** |
| H1: There is no change in the water cognition of the public in the crisis of water shortage. | | |
| Water use awareness | Drinking habit at home | 0.74 |
| | Drinking habits when going out | 0.75 |
| | Drink bottled water per week | 0.73 |
| | Boiling water time in bath | 0.74 |
| | The look and feel of tap water | 0.74 |
| | The look and feel of bottled water | 0.74 |
| H2: There is no change in the water attitude of the public in the crisis of water shortage. | | |
| Water attitude | Good tap water quality | 0.72 |
| | Tap water can be consumed directly | 0.73 |
| | Chlorinated tap water is safer | 0.73 |
| | The water price of the residential area is reasonable | 0.73 |
| | Taiwan's water price is reasonable, no sense of water consumption | 0.73 |
| | The people of Taiwan have no sense of water crisis | 0.72 |
| | Residents have no awareness of water crisis | 0.73 |
| | Willing to save water | 0.71 |
| | Bottled water will cause environmental load | 0.71 |
| | Willing to bring a water bottle instead of plastic bottled water | 0.71 |
| | The quality of tap water is better than bottled water | 0.72 |
| | Clean and safe water delivery to home is a simple matter | 0.72 |
| | The government should bear the greatest responsibility for water conservation | 0.71 |
| H3: There is no change in the water behavior of the public in the crisis of water shortage. | | |
| Water use behavior | Will turn off the head at will | 0.88 |
| | Will pay attention to the time it takes to take a bath | 0.87 |
| | Clean the water tower of the pool at home at least once a year | 0.88 |
| | I always try to bring my own eco-friendly cup | 0.87 |
| | Will choose environmentally friendly cleaning products | 0.87 |
| | Will buy products with "water mark" | 0.86 |
| | When water leaks are found, relevant units will be proactively notified | 0.87 |
| | Will add Jingxi activities | 0.86 |
| | Will add beach cleaning activities | 0.86 |
| | Will add clean sea activities | 0.86 |
| | Will pick up rubbish by the creek or lake bank | 0.86 |

However, national public health issues need to be examined in a more cautious and well-developed manner, and more accurate and in-depth information can be obtained by using a hybrid research method to collect additional information in a variety of ways for comparison or corroboration [58–64]. Therefore, after obtaining the data from the questionnaire sample, the study used video software or a telephone to interview students,

housewives, swimming pool operators, school teachers, as well as experts and scholars to consolidate the information in a rigorous sequence of compilation, organization, and analysis to construct the final article [53], which was finally analyzed and explored by multivariate validation [54,55].

### 2.3. Study Scope and Limitations

The study was conducted to investigate the water literacy of the public in Taiwan. Due to the threat of the COVID-19 epidemic during the sampling period, and the financial, human, and material constraints, the finalized questionnaire on water awareness, attitudes, and behaviors was administered through a web-based questionnaire platform, using the intentional sampling method to seek out people in each region to be interviewed, and then requesting the assistance of people who had already been surveyed, using the snowball sampling method to help expand the scope of the sample and the target population. After collection and analysis, interviews were conducted using a video conferencing system and telephone after obtaining respondents' consent to be interviewed, and finally, a multivariate verification method was used.

The sampling method and sample size were affected by the limitations of funding, human and material resources, and the epidemiological environment of this study. The methodologies used in this study might have caused slight differences in the inferences and analysis results. Although the sample size of this study was 653, such a sample size has achieved a 5% sampling error and 95% confidence level, which is sufficient to represent a population size of more than 100,000 [55,56]. In addition, the combination with a qualitative approach complemented the shortcomings of the present study [65]. However, the sample size may still be adjusted in order to obtain more accurate results. Due to the financial, time, and physical constraints of the researcher and the urgency of the current water supply restriction situation, only 653 questionnaires were used for inference and analysis. The inferred results can only represent the views of the respective sampled subjects and may differ from the views of the unsampled population. Therefore, the shortcomings of the relevant studies will be listed in future research recommendations for enhancement and corroboration by subsequent researchers.

### 2.4. Ethical Considerations

The data for this study were collected from residents in northern, central, southern, eastern, and outlying islands of Taiwan by intentional sampling. During the data collection process, Taiwan was still facing the COVID-19 crisis, so going out to interview was risky. In addition, the lack of water in Taiwan had increased the stress of people's lives and work, making them less willing to be interviewed. The snowball sampling method is characterized by low time and personnel costs and accurate and effective data collection [66]. Therefore, the questionnaire was distributed outwardly around the research team and the members involved. Utilizing the functionality and convenience of the web-based questionnaire platform, 653 valid data were eventually obtained for analysis from respondents who indicated their willingness to be interviewed after fully understanding the topic and significance of the study [66]. Therefore, all respondents agreed to provide the relevant data with the understanding of the study's main purpose. All questionnaires and interviews were recorded and collected from anonymous and informed respondents.

### 3. Analysis of Results

A total of 653 questionnaires were collected, and the background of the respondents was analyzed using statistical tests. Then, we analyzed the cognition, attitudes, and behaviors of people in different regions and the related effects by statistical analysis, ANOVA, and Pierson product–moment correlation analysis. The analysis showed that the majority of the respondents were male (66.9%) and the least were female (33.1%); the largest number of respondents were aged 21–30 (25.9%) and the least were aged 61 or older

(1.1%); the majority of respondents live in the central region (41.7%) and the least in the east (1.1%), as shown in Figure 2.

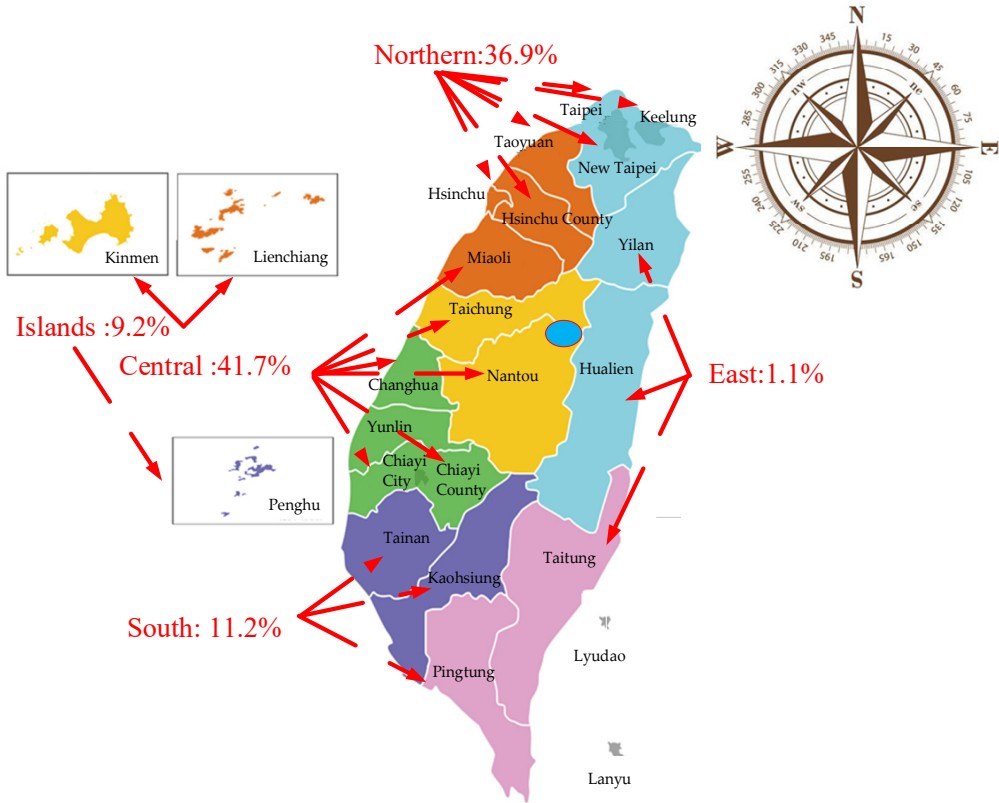

**Figure 2.** Distribution of sample population.

### 3.1. Water Use Awareness Analysis of the Taiwanese Public

With climate change causing freshwater regulation failures, island countries are facing water shortage crises, and Taiwan is facing the same problem [11,12]. Understanding the public's awareness of water literacy [22], finding a cooperative solution to the water shortage crisis, and sustaining water resources will be some of the simplest and most direct responses [45].

The analysis of people's water literacy awareness revealed that in terms of drinking habits at home, boiling and filtering was the highest (38.1%) while drinking directly (1.8%) was the lowest. In terms of drinking habits outside the home, the highest awareness was for bringing water from home (40.6%) and the lowest was for drinking water directly from the tap (0.5%). In terms of the number of bottled waters consumed per week, more than six bottles (34.6%) was the highest, and not drinking bottled water (13.2%) was the lowest. In terms of the duration of water running in the shower, 6–15 min (56%) was the highest and 26 min or more (2.3%) was the lowest. In terms of the perception of tap water, more convenient (45%) was the highest, and better taste (3.7%) was the lowest. In terms of the perception of bottled water, more convenient (53.3%) was the highest, and healthier (3.5%) was the lowest.

There were significant differences ($p < 0.05$) in drinking habits (2.18:2.02) and perceptions of tap water (3.54:3.92) among different genders, as well as among different ages. Significant differences ($p < 0.05$) were found in the number of bottled waters consumed per week, the duration of water running during bathing, perception of tap water, and perception of bottled water among people in different regions, and people in the east were more sensitive to the perception of tap water, while people in the outlying islands were more sensitive to the perception of bottled water. As shown in Table 2.

**Table 2.** Water use awareness analysis of the Taiwanese public.

| Issue | Lowest | Highest | Gender | Age | Place of Residence |
|---|---|---|---|---|---|
| | | | (Male:Female) | Levene (F) | Levene (F) |
| Drinking habit at home | Direct drinking (1.8%) | Boiling and filtering before drinking (38.1%) | (3.50:3.66) | (0.000) | (0.000) |
| Drinking habits when going out | Drinking water directly from the tap (0.5%) | Bringing water from home (40.6%) | (2.18:2.02) * | (0.119) * | (0.661) |
| Drink bottled water per week | Not drinking bottled water (13.2%) | More than six bottles (34.6%) | (2.83:2.69) | (0.010) | (0.164) * |
| Boiling water time in bath | 26 min or more (2.3%) | 6–15 min (56%) | (1.69:1.85) | (0.050) | (0.220) * |
| The look and feel of tap water | Better taste (3.7%) | More convenient (45%) | (3.54:3.92) * | (0.000) * | (0.082) * (Eastern > Northern, Central, Southern) |
| The look and feel of bottled water | Healthier (3.5%) | More convenient (53.3%) | (4.10:4.08) | (0.002) | (0.235) * (Outlying Islands > Northern, Central) |

* $p < 0.001$.

The results of the analysis are shown in Figure 3. Understanding people's awareness of water literacy [22] and seeking public consensus to work together to solve the water shortage crisis will help to sustain water resources and solve the water shortage crisis [45]. From the above inferences, the investigators concluded that well-educated people have higher health and hygiene knowledge, are sophisticated and cautious about household water treatment measures, and are generally aware of the concept that filtered and boiled water can kill bacteria. Therefore, people considered tap water to be more convenient (45%) and would boil and filter it before drinking (38.1%), and thought it tasted good, but would not drink it directly.

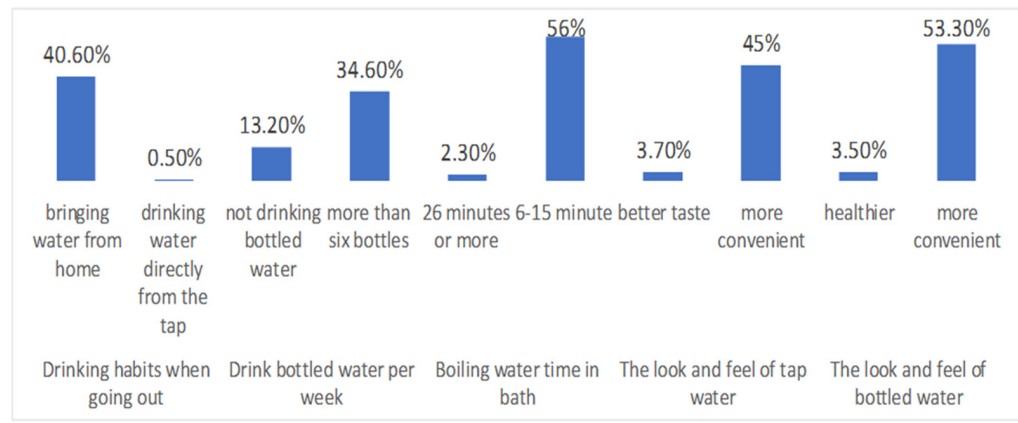

**Figure 3.** Water use awareness analysis.

Even so, most people are busy coping with work, school, and family care due to the fast pace of work and life in a society with advanced technology and highly developed industry and commerce Very few people are able to prepare water for work before they leave. As a result, people drank more than six cans of bottled water per week (34.6%). Only a very small percentage of people did not buy bottled water to drink (13.2%).

Yet, the overall environment is beginning to change. With the poor economy, serious air pollution, and concerns about plasticizers that may be released by plastic water bottles, people are more cautious about water use. Therefore, although people considered bottled water to be more convenient (53.3%), in terms of drinking habits when they were away from home, most people chose to bring water from home (40.6%), and the fewest people drank water directly from the tap (0.5%) and did not consider tap water to be healthier (3.5%).

For a long time, the government has been promoting the policy of water conservation at all levels of education or departmental planning. The concept of installing water-saving devices on the water pipes in order to reduce water costs has been deeply rooted in the

lives and awareness of the public. As a result, most interviewees stayed within 6 to 15 min of bathing time (56%), and few took a bath for more than 26 min (2.3%).

### 3.1.1. Water Use Attitude Analysis of the Taiwanese Public

Enhancing individuals' attitudes toward the importance of water conservation [45,46] can help increase people's awareness of the sustainability, maintenance, and importance of water [22,47].

The analysis found that, in terms of people's attitude towards water literacy, awareness was highest for the willingness to carry a water bottle instead of plastic bottled water (4.32), good tap water quality (3.36), and that the government should take the greatest responsibility for water conservation (3.71), and lowest for tap water being directly drinkable (1.90), the price of water in Taiwan being cheap and water consumption being insensitive (2.80), and the delivery of clean, safe tap water to the home being a simple task (2.72). Significant differences ($p < 0.05$) were found between genders in the perceptions of good tap water quality (3.40:3.27), chlorinated tap water being safer (2.54:2.62), no awareness of a water crisis (3.46:3.69), willingness to save water (4.29:4.25), and willingness to carry a water bottle instead of plastic bottled water (4.28:4.38). There was a significant difference ($p < 0.05$) in the opinion that chlorinated tap water is safer by age, while other opinions were consistent. Significant differences were found between regions ($p < 0.05$) on the issues of drinking water directly, the price of water in Taiwan, senseless water consumption, and willingness to conserve water, with people in the south feeling more strongly than those in the north and central regions on the issue of drinking water directly. As shown in Table 3.

**Table 3.** Water use attitude analysis of the Taiwanese public.

| Issue | | M | Rank | Gender (Male:Female) | Age Levene (F) | Place of Residence Levene (F) |
|---|---|---|---|---|---|---|
| Water perception | Good tap water quality | 3.36 | 1 | (3.40:3.27) * | (0.015) | (0.674) |
| | Tap water can be consumed directly | 1.90 | 3 | (2.02:1.66) | (0.340) | (0.024) * (Southern > Northern, Central) |
| | Chlorinated tap water is safer | 2.57 | 2 | (2.54:2.62) * | (0.043) * | (0.090) |
| Water crisis | The water price of the residential area is reasonable. | 2.96 | 5 | (3.01:2.86) | (0.508) | (0.687) |
| | Taiwan's water price is reasonable, no sense of water consumption | 2.80 | 6 | (2.83:2.74) | (0.031) | (0.509) * |
| | The people of Taiwan have no sense of water crisis | 3.54 | 4 | (3.46:3.69) * | (0.004) | (0.138) |
| | Residents have no awareness of water crisis | 2.96 | 5 | (2.97:2.94) | (0.001) | (0.057) |
| | Willing to save water | 4.28 | 2 | (4.29:4.25) * | (0.028) | (0.087) * |
| | Bottled water will cause environmental load | 4.13 | 3 | (4.10:4.20) | (0.004) | (0.149) |
| | Willing to bring a water bottle instead of plastic bottled water | 4.32 | 1 | (4.28:4.38) * | (0.052) | (0.358) |
| Responsibility | The quality of tap water is better than bottled water | 2.92 | 2 | (2.91:2.94) | (0.026) | (0.350) |
| | Clean and safe water delivery to home is a simple matter | 2.72 | 3 | (2.74:2.69) | (0.012) | (0.185) |
| | The government should bear the greatest responsibility for water conservation | 3.71 | 1 | (3.67:3.79) | (0.000) | (0.026) |

\* $p < 0.001$.

The overall results of the analysis are shown in Figure 4. Establishing people's attitudes toward the conservation of freshwater resources [45,46] helps to enhance the awareness of maintaining the water environment and sustainable water resources [22,47]. Years of environmental education and the government's fine water quality control for household use have been well received by the public. Yet, the water shortage crisis cannot be solved overnight. In the face of the water shortage crisis, the government has announced the regulation of irrigation water, implemented artificial rainfall, and even issued a document encouraging Taiwan's water-related leisure industry to voluntarily suspend operation in order to conserve water. Taiwan's reservoirs have been heavily silted up [67], and their storage capacity is obviously reduced [7,11,12]. However, the government has not seized the opportunity to dredge reservoirs, repair reservoir storage facilities, or maintain the

environment of lakes during the dry season. Furthermore, the government's main message is to ask people to conserve water, adjust the timing of agricultural irrigation, and ask water industry owners to shut down their businesses voluntarily, but no water restriction measures have been implemented for industrial and commercial users of fresh water [68]. Consequently, most people considered that Taiwan's tap water was of good quality (3.36) but difficult to deliver (2.72) and expensive (2.80). People would not drink the water directly (1.90) and would prefer to carry a water bottle instead of plastic bottles (4.32). They already felt the water shortage crisis (2.80) and believed that the government should be the most responsible for the current crisis of water shortage in Taiwan (3.71).

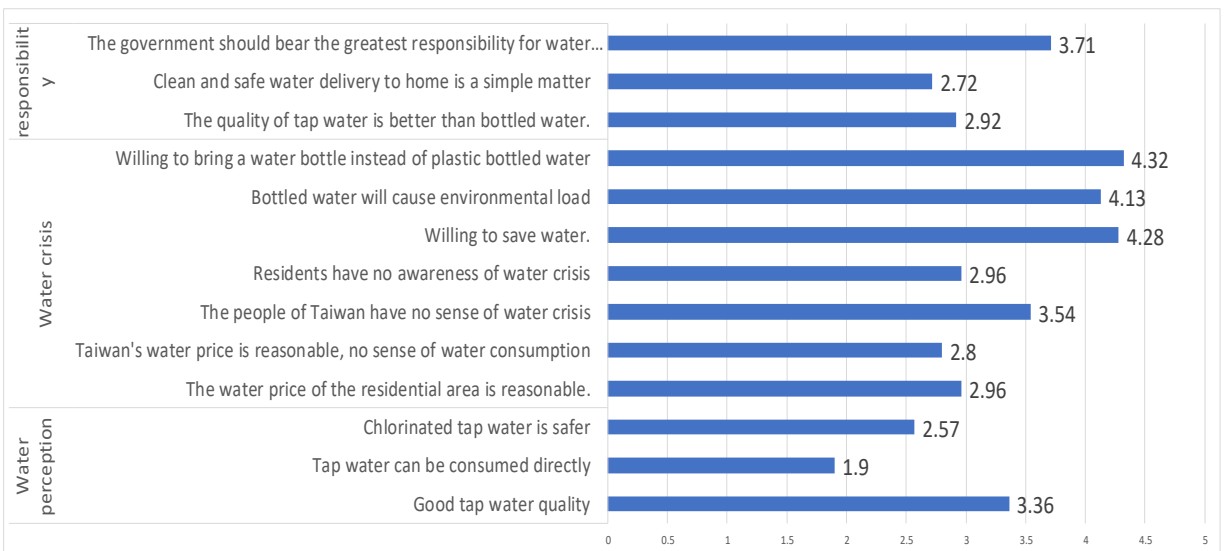

**Figure 4.** Analysis of water use attitude.

The overall analysis results are shown in Table 4. Taiwan has good control over water resources and sanitation quality [69,70], and the quality of life is high. However, being an island country surrounded by sea with short waterways that flow fast, Taiwan has difficulty in storing fresh water [7,11,12]. Although people are facing a water shortage crisis and their work will be affected by the water limitation, they are more concerned about the safety and the health issue of drinking water at the moment. Moreover, since women are more concerned about health and most of them have family management responsibilities, men are more concerned about issues such as tap water quality and the awareness of a water crisis among Taiwanese people, while women are more concerned about issues such as tap water safety, water conservation, and carrying water bottles instead of plastic bottled water. People of different ages also have different views on issues such as chlorinated tap water being safer.

Although reservoirs have been built to store water in all regions of Taiwan, and prudent and complete measures are in place for sourcing, storing, disinfecting, and precipitating domestic water, the intensive industrialization and frequent thermal power generation in the northern half of Taiwan have affected public perceptions of water safety and sanitation in the region and raised concerns. In addition, there are obvious differences in the measures and effectiveness of water storage and freshwater transport between the east and west, resulting in different perceptions of water consumption and the convenience of using household water. As a result, respondents in different regions have different views on issues such as water prices, drinking tap water, freshwater consumption, and willingness to save water. In particular, with regard to the issue of the direct drinking of tap water, respondents in the southern region felt more strongly than those in the northern and central regions.

**Table 4.** Water use behavior analysis of the Taiwanese public.

| | Issue | M | Rank | Gender | Age | Place of Residence |
|---|---|---|---|---|---|---|
| | | | | (Male:Female) | Levene (F) | Levene (F) |
| Ecological Management Action | Will turn off the head at will | 4.60 | 1 | (4.57:4.65) * | (0.000) | (0.000) |
| | Will pay attention to the time it takes to take a bath | 4.20 | 3 | (4.17:4.24) | (0.292) * (41–50 > 20 down; 51–60 > 20 down; 31–40) | (0.000) |
| | Clean the water tower of the pool at home at least once a year | 3.78 | 4 | (3.77:3.81) | (0.018) * (20 down < 31–40; 41–50; 51–60; 51–60 > 21–30) | (0.136) * (Northern, Central, Southern > Outlying Islands) |
| | I always try to bring my own eco-friendly cup | 4.21 | 2 | (4.11:4.40) | (0.950) * (20 down < 41–50; 51–60; 51–60 > 21–30) | (0.825) * (Central, Southern > Outlying Islands) |
| Consumer Economic Action | Will choose environmentally friendly cleaning products | 4.08 | 1 | (4.05:4.15) * | (0.360) (20 down < 41–50; 51–60; 51–60 > 21–30; 31–40) | (0.084) * (Northern, Central, Southern > Outlying Islands) |
| | Will buy products with "water mark" | 4.05 | 2 | (4.01:4.15) | (0.027) * (31–40; 51–60; 61 up > 20 down) | (0.012) * (Northern, Central, Southern > Outlying Islands) |
| Citizen action | When water leaks are found, relevant units will be proactively notified | 4.11 | 1 | (4.12:4.08) | (0.445) * (31–40; 41–50; 51–60; 61 up > 20 down) | (0.091) * (Northern, Central, Southern > Eastern) |
| | Will add Jingxi activities | 3.57 | 5 | (3.60:3.52) | (0.000) | (0.004) |
| | Will add beach cleaning activities | 3.66 | 3 | (3.66:3.66) | (0.001) | (0.420) |
| | Will add clean sea activities | 3.62 | 4 | (3.62:3.63) * | (0.000) | (0.260) |
| | Will pick up rubbish by the creek or lake bank | 3.80 | 2 | (3.843.73) | (0.013) * (61 up > 20 down) | (0.571) |

* $p < 0.05$.

### 3.1.2. Water Use Behavior Analysis of the Taiwanese Public

Increasing the public's recognition of sustainable water resources can help achieve the goal of water conservation by promoting spontaneous conservation measures or actions [48].

The analysis found that, in terms of public awareness of water use, the ecological management action of turning off the tap (4.60) was the highest, while the lowest was to clean the water tower at home at least once (3.78). In terms of consumer economic action, the highest was to choose environmentally friendly cleaning products (4.08), while the lowest was to buy products with a "water proficiency label" (4.05). In terms of civic action, the percentage of those who take the initiative to report water leakage to the relevant authorities (4.11) was the highest, while the percentage of those who participated in stream-cleaning activities (3.57) was the lowest.

Significant differences ($p < 0.05$) were found among different ages in the issues of paying attention to the time spent on bathing, cleaning the water tower at home at least once, bringing their own eco-cups as much as possible, choosing eco-friendly cleaning products, purchasing products with the "water proficiency label", taking the initiative to report water leakage to the relevant authorities, and picking up trash at the stream or lakeshore, etc. The awareness of paying attention to the time spent on bathing was high among people aged 41–60. People aged 20 and younger and 51–60 were the least conscious of cleaning the water tower at home at least once and choosing environmentally friendly cleaning products; people aged 31–40, 51–60, and 61 and older agreed the most when purchasing products with the "water proficiency label"; people aged 31–40, 41–50, 51–60, and 61 and older were the most proactive in reporting water leakage to the relevant authorities; people aged 61 and older were the most active in picking up trash at the stream or lakeshore.

There were significant differences ($p < 0.05$) among regions in terms of cleaning the water towers at home at least once, bringing their own eco-cups as much as possible, choosing eco-friendly cleaning products, and taking the initiative to report water leakage to the relevant authorities upon discovery. People in the north, central, and south were more cooperative than those in the east in purchasing products with the "water proficiency label" and taking the initiative to report water leaks to the relevant authorities. As shown in Table 4.

The goal of water conservation can be achieved by encouraging people to take conservation measures or actions on their own initiative [48].

In Taiwan, water conservation education has been effectively implemented, and people have developed sufficient energy-saving and water-saving habits. However, most people are reluctant to go to the water towers to clean the dirt themselves due to factors

such as the constraints of work and pressure, high locations of water towers, and limited mobility. Therefore, in terms of environmental management actions, the majority of people are willing to turn off the tap (4.60), but they are least willing to clean the water tower at home at least once (3.78).

The public has a wide range of consumer goods to choose from, but the increased awareness of ecological and environmental conservation will inevitably peak consumers' interest to purchase goods that are beneficial to the environment. However, the current high unit price of products with water proficiency labels may affect consumers' willingness to purchase them. Therefore, regarding the economic actions of the public consumers, the highest number of people chose to buy eco-friendly cleaning products (4.08), but the number of people who bought products with the " water proficiency labels" was lower (4.05).

Although Taiwan's freshwater resources are easily accessible, they are not enough. The public's awareness of water resources is deep rooted, helping people to take actions to protect water resources. However, nature conservation takes time. Long rivers and streams are not easily accessible and, therefore, are time-consuming to clean up, which affects the public's willingness to clean up streams. Therefore, the willingness of citizens to take the initiative to report water leakage to the relevant authorities was the highest (4.11) and to participate in stream-cleaning activities was the lowest (3.57). As shown in Figure 5.

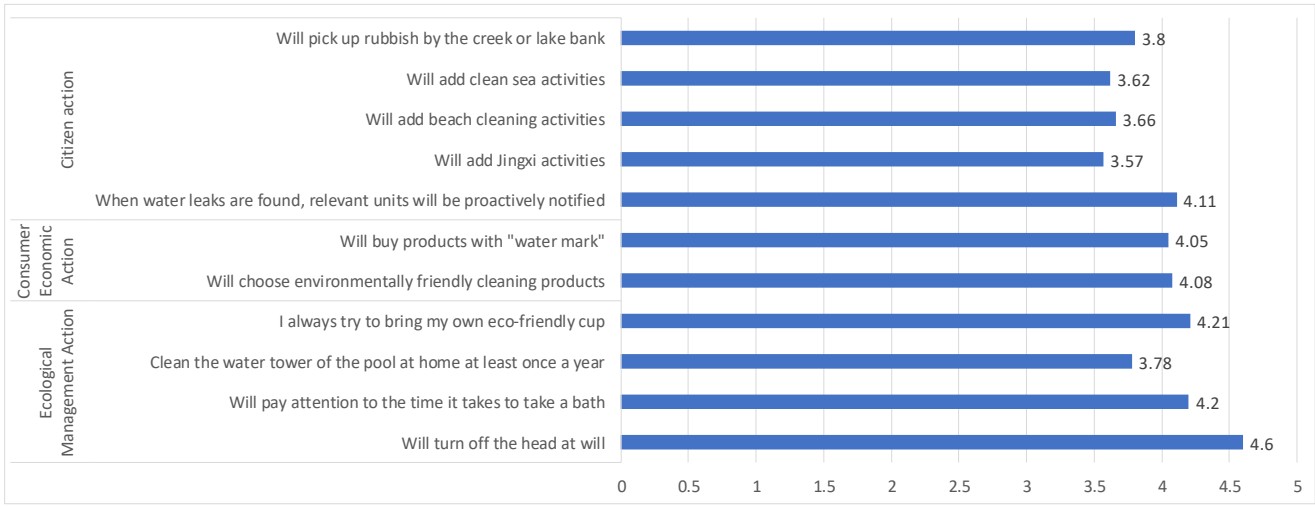

**Figure 5.** Analysis of water use behavior.

Based on the number of samples obtained in the research, the opinions of respondents in different regions are analyzed. People of different ages have different requirements in terms of mobility and time to deal with the bathing process. In addition, factors such as time pressure at work or school and spending power may limit people's willingness to consume environmentally friendly products at high unit prices, thus influencing people's civic behavior. As a result, people of different ages have different opinions on issues such as paying attention to the time spent on bathing, cleaning the water tower at home at least once a year, bringing their own eco-cups as frequently as possible, choosing eco-friendly cleaning products, buying products with the "water proficiency labels", taking the initiative to report water leakage to the relevant authorities, and picking up trash at the stream or lakeshore.

Most people aged 41–60 have already started a family and have the responsibility and pressure of taking care of the family, so they attach more importance to the concept of water conservation. People aged under 20 or between 51 and 60 had low intention to clean the water tower due to work pressure and reduced mobility. Those aged 31–40, 51–60, or more than 61 had a stable financial ability, so they were willing to buy "water proficiency label" products. Most people aged 61 and above were facing retirement and had more

time available. Therefore, people aged 41–60 were more concerned about the time spent on bathing, while people under 20 and aged between 51 and 60 were more reluctant to clean the water towers in their homes. People aged 31–40, 51–60, or 61 and higher had the greatest consensus on buying products with "water proficiency labels". People aged 31–40, 41–50, 51–60, and 61 years old would take the initiative to notify the relevant authorities if they discovered a water spill, and it was most common for people aged 61 and above to pick up trash along the stream or lakeshore.

Although Taiwan has good control of water resources, the western half of the country has been suffering from poor air quality and excessive air particles due to climate change, the northeastern monsoon, and the current use of thermal power as the main source of electricity generation [71]. In addition, there are differences in infrastructures, consumption abilities, and living standards between different regions. As a result, respondents from different regions disagreed on behaviors such as cleaning the water tower at home at least once a year, carrying an eco-friendly cup, choosing eco-friendly cleaning products, and taking the initiative to notify the relevant authorities when water resources leak.

In the north, central, and south, most of the buildings are condominiums or high-rise buildings with towering heights and dense construction. Water tower facilities are usually located at high points, making them difficult to access. Air pollution has been a serious problem for many years, and people became increasingly conscious of personal safety and health care. Hence, respondents living in the west indicated that they would buy cleaning products that would contribute to environmental protection and would be willing to improve the quality of water used for domestic purposes. Respondents in the north, central and south are more willing to clean their water towers once a year and choose environmentally friendly cleaning products than those in the outlying islands.

However, people in the western half of the country have plenty of amenities, more job opportunities, and higher incomes than those in the eastern half. In addition, the western half of the country has a large population, a dense workforce, and a high demand for water, so they are more willing to report water leaks if found. Therefore, people in the northern, central, and southern parts of the country were more willing to buy products with the "water proficiency labels" than people in the eastern part of the country and would take the initiative to report water leaks to the relevant authorities when they were discovered.

### 3.1.3. Relevant Analyses of Water Use Awareness, Attitudes, and Behaviors of the Taiwanese Public

It is known from the literature that establishing good water literacy helps people to understand the importance of water conservation actions and good management, to recognize and take certain actions to preserve water resources and establish their value, and to communicate relevant messages effectively and clearly [37], which helps to enhance the correct use of water resources [22].

The analysis revealed a low significant effect ($p < 0.01$) between water use attitude and water use behavior (0.361) and a low significant effect ($p < 0.01$) between water use awareness (0.29) and crisis (0.11) and civic action. There was a low significant effect ($p < 0.01$) among water responsibility and ecological management actions (0.227), consumer economic actions (0.200), and civic actions (0.115). No significant effect was found for water use awareness, attitude, and behavior ($p > 0.05$). As shown in Table 5.

**Table 5.** Relevant analyses of water use awareness, attitudes, and behaviors of the Taiwanese public.

| | Water Use Awareness | Water Use Behavior | Ecological Management Action | Consumer Economic Action | Citizen Action |
|---|---|---|---|---|---|
| Water use awareness | 1 | 0.047 | 0.061 | −0.020 | 0.046 |
| Water attitude | −0.031 | 0.361 ** | 0.344 ** | 0.288 ** | 0.289 ** |
| Water perception | −0.025 | 0.077 | 0.064 | 0.064 | 0.110 ** |
| Water crisis | −0.033 | 0.231 ** | 0.227 ** | 0.200 ** | 0.115 ** |
| Water responsibility | −0.008 | 0.266 ** | 0.249 ** | 0.188 ** | 0.309 ** |

** $p < 0.001$.

The consolidation of the conservation mindset helps to build the pre-conscious behavior of the public and promote the generation of issue-specific spontaneous actions [65]. The establishment of mindsets toward water use will help to improve people's attitudes toward the conservation of freshwater resources [45,46] and enhance the effectiveness of water conservation [22,47]. Therefore, when people's attitudes toward water use increase and they consider the conservation of freshwater resources as their own responsibility, environmental management actions, consumer economic actions, and civic actions will be promoted. When the sense of water perception and crisis is strengthened, the awareness of civic actions will also be enhanced. Therefore, there was a slight correlation between water use attitudes and water use behaviors, while water use responsibility was significantly correlated with environmental management actions, consumer economic actions, and civic actions. There was also a correlation between water use awareness and crisis and civic actions.

## 4. Discussion

In this work, based on the case of Taiwan, a total of 653 questionnaires were used for inference and analysis to explore the public's knowledge, attitude, and behavior regarding water in order to provide current and future countermeasures against water scarcity.

### 4.1. Water Use Awareness Analysis

We believe that Taiwan has low water prices, good management, and widespread water literacy awareness and behavior. Therefore, most people are willing to reduce water waste and pay attention to the hygiene and safety of drinking water. However, most people still choose to buy bottled water to drink due to the fast pace of work and life, the convenience of consumption, and the long boiling time for the sterilization of drinking water.

However, due to differences in personality, lifestyle, and nature of work occupation, men seek the convenience of drinking water, while women choose their sources carefully. In addition, there are differences in water sources, water quality, and water needs between regions, as well as differences in living habits, work pace, ease of consumption, tourism development, and environmental maintenance. Therefore, respondents living in the eastern part of the country have a higher demand for high-quality household water. Respondents in the outlying islands were more sensitive to the use of bottled water. The choice of drinking water source and knowledge of water use behavior also varied by region.

It is recommended to install stable and hygienic public drinking water in public areas, to improve the quality of water sources in the eastern part of the island and to increase the effectiveness of environmental maintenance for outlying island tourism. The use of bottled water should be reduced by promoting the use of eco-cups. The above will help improve the current state of public water awareness and the dilemma of gender differences in water awareness.

### 4.2. Water Use Attitude Analysis

We believe that, although Taiwan is currently facing a drought and has taken measures such as artificial rainfall, zoning irrigation, and restricting industrial water use, reservoirs are heavily silted up and the amount of water stored has decreased. The drought coupled with heavy water use has reduced the amount of water in the existing reservoirs and increased the turbidity of the water sources, which has affected the quality of the water sources. As a result, most people are skeptical of the quality of water sources and do not trust the government's water management decisions and effectiveness.

However, due to differences in personality, life and work demands, family upbringing, and social responsibilities, men are more concerned with social issues, while women are concerned with household water safety and sanitation. In addition, thermal power generation is the main source of electricity supply in Taiwan. Due to topographical and climatic factors, dust from power generation drifts to the central and southern regions, affecting the

environment, polluting the air, and indirectly affecting water sources. Although there are many reservoirs in the central and southern regions with a large water storage capacity, it still affects the public perception. Therefore, respondents living in the southern region are particularly concerned about water quality.

It is recommended that: the government should reformulate its water management policies and implement water management measures; improve dust pollution from thermal power generation; plan urban water storage spaces and design urban water-recycling measures to reduce public water use; and increase the government's visibility by improving the actual effectiveness of its policies. Through these measures, positive public perceptions of government water management decisions can be increased and differences in attitudes toward water use by gender and place of residence can be improved.

*4.3. Water Use Behavior Analysis*

We conclude that Taiwanese people are accustomed to convenient water supply measures and that the high location of household water storage facilities makes cleaning difficult. The wide range of environments in water source areas requires a long maintenance time. In addition, the unit price of commercially available water-saving facilities or products is high. These factors make most people much less willing to maintain the environment of the water source area, neglect water conservation behaviors, and reluctant to purchase water conservation facilities.

Although water-saving behaviors such as cleaning and maintenance are less convenient and related facilities and products are expensive, water literacy education has been established in Taiwan, and water conservation can help reduce the financial burden on households. Thus, it can attract retirees aged 61 and older to participate in environmental cleanup initiatives and promote water conservation among those aged 41–60. However, it does not increase the willingness of people aged 31–40, 51–60, and 61 and older to purchase water-saving products, nor does it increase the willingness of students under 20 to participate in environmental activities. In addition, there are differences in geography, climate, and current air and environmental pollution issues in different regions. However, due to the high population density, high consumption levels, and mostly high-rise buildings in the western, cleaning companies are generally available to maintain the environment of the buildings and water storage tanks. Therefore, compared to respondents in the east, people in the north, central and south are more willing to buy products with the "water proficiency labels" and are willing to take the initiative to report water leaks so that they can be fixed.

It is recommended that environmental groups be organized to raise public awareness of environmental protection. The government should also promote the establishment of cleaning companies for water storage facilities to help people maintain the water environment and increase employment opportunities. By investing in public facilities or subsidizing water-saving devices and products in residential buildings, the government can effectively achieve water-saving actions, raise public awareness of water use behavior, and improve the differences in water use behavior between genders and places of residence.

*4.4. Analysis of the Correlation Between Water Use Awareness, Attitude, and Behavior*

We concluded that although building water knowledge can help reduce internal consumption of water resources, attitudes toward adequate water use remain limited due to economic pressures, lack of time, and limited mobility. As a result, even though a correlation exists between water responsibility and ecological management actions, consumer economic actions, and citizen actions, as well as between water perceptions and crises and citizen actions, the impact is small.

It is recommended that the government invest in energy-efficient water facilities in public areas, develop urban water conservation measures, and subsidize household water-saving devices so that people can make good use of their learned water awareness and positive attitudes toward water use to achieve their water conservation goals.

## 5. Conclusions and Recommendations

According to the results of the analysis, people generally have a high level of water knowledge and have developed sufficient water conservation habits. Most people believe that tap water is of good quality, but it is difficult to deliver, expensive, and should not be consumed directly. Although they already sense the crisis of water shortage, they are willing to carry water bottles instead of plastic bottled water, choose to buy environmentally friendly cleaning products, perform water conservation behaviors on the go, limit showers to 6–15 min, and take the initiative to notify relevant units to repair water facilities. However, due to constraints such as working and daily habits, consumption ability, and mobility, people still use more bottled water, are less willing to buy products with the "water proficiency labels" and participate in stream-cleaning activities, and believe that the government is most responsible for Taiwan's current water shortage crisis. In addition, increasing water responsibility can help enhance environmental management actions, consumer economic actions, and civic actions, and enhance the perception of water.

Different genders have different views on drinking habits, perception of tap water, tap water quality, Taiwanese people's awareness of the water crisis, tap water safety, water conservation, and bringing water bottles instead of plastic bottles, due to their work needs and life roles. People of different ages, depending on their jobs, education, and mobility, have different opinions on drinking habits outside, perception of tap water, chlorinated tap water being safer, paying attention to the time spent on bathing, cleaning the water tower at home at least once a year, bringing their own eco-cups as much as possible, choosing eco-friendly cleaning products, buying products with the "water proficiency labels", taking the initiative to report water leakage to the relevant authorities, and picking up trash at the stream or lakeshore. The number of bottled waters consumed per week, the time to turn on the water for bathing, the perception of tap water, the perception of bottled water, the price of tap water, freshwater consumption, water conservation, cleaning the water tower at home at least once a year, bringing one's own eco-cup, choosing eco-friendly cleaning products, and taking the initiative to report water leakage to the relevant authorities vary from region to region depending on the level of living and consumption, and the convenience of living.

Based on the above findings, the following recommendations are made:

(1) For the government and related organizations It is recommended that the government should not only continue to improve and provide subsidies to enhance the existing water resources technology and improve the development and design technology, but also continue to track the weather and implement artificial rainfall measures, and immediately carry out the dredging of rivers and reservoirs, and actively negotiate with neighboring countries to prepare for the purchase of water in order to cope with the current water shortage.

(2) For water-related companies or organizations It is recommended that water-related organizations or institutions should make immediate water conservation decisions during current water shortages. The current problem can be solved if water-saving or water-recycling facilities are used without affecting the industrial movement.

(3) For the public Although bottled water is convenient, people still need to consider the environmental hazards that come with its use. It is recommended that people still carry environmentally friendly cups to reduce consumption and increase environmental literacy when traveling to streams, rivers and other water areas to reduce the problem of trash left behind in order to reduce the frequency of cleaning streams or beaches or the risk of doing so.

(4) For other countries The present study found that the quality of domestic water sanitation control in Taiwan is good, and people still have a consensus and behavior to conserve water, but the practical actions they can take are limited due to time, economic, and safety considerations. Thus, water scarcity will be a common problem in the future for countries suffering from climate anomalies and limited freshwater resources, and facing economic and human development needs. Therefore, it is recom-

mended to implement water literacy awareness for citizens, promote environmental knowledge and friendly environmental behavior, and encourage the funding and establishment of environmental groups to help maintain the community's environment. These actions will help governments and people in other countries and regions to overcome the water shortage crisis in the future.

(5) Suggestions for future studies Due to the limitations of this study in terms of funding, human and material resources, and the epidemic setting, the sampling method and questionnaire size were compromised, which affects the accuracy of the study's conclusions. It is recommended that subsequent researchers expand the sample size of the study and include other research methods and subjects for study. Subsequent studies can be conducted in other regions or countries to fill in the gaps. Furthermore, water resources are an important issue, and there are many different ways to obtain water, so the design of convenient and evaluated water collection facilities for domestic use will be the goal of future development.

**Author Contributions:** Conceptualization, J.-H.Y. and H.-H.L.; methodology, J.-H.Y.; software, J.-H.Y.; validation, Y.-C.L., H.-H.L. and C.-H.H.; formal analysis, K.-C.T.; investigation, J.-H.Y.; resources, Y.-C.L.; data curation, C.-H.H.; writing—original draft preparation, C.-H.H.; writing—review and editing, H.-H.L.; visualization, H.-H.L.; supervision, Y.-C.L.; project administration, K.-C.T.; funding acquisition, K.-C.T. All authors have read and agreed to the published version of the manuscript.

**Funding:** This research received no external funding.

**Institutional Review Board Statement:** Not applicable.

**Informed Consent Statement:** Not applicable.

**Data Availability Statement:** No data provided.

**Acknowledgments:** We thank all the interviewees who participated in the research.

**Conflicts of Interest:** No conflicts of interest.

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
