# Peer review of "Measures to Cope with the Impact of Climate Change and Drought in the Island Region: A Study of the Water Literacy Awareness, Attitude, and Behavior of the Taiwanese Public"

_water, doi:10.3390/w13131799_

Round 1
Reviewer 1 Report
As I mentioned in the previous review, the Introduction provides very detailed and expressive data based upon keen research and study. The innovative aspects are well highlighted, full of content, supported by relevant and up to date references. I noticed that the chapter Introduction is more developed in this improved version.
Methods and Instruments point out in a professional manner the undertaken research, bringing a very clear view on the importance of education related to water literacy and awareness including the innovative questionnaire, which is a very good tool to extract both the issue and the solution. I noticed that figure 1 is improved, being much more clearly illustrated the structure of the research.
Unlike the previous version, this manuscript also presents a map from where the questionnaires were collected, which clarifies the distribution of sample population. The Analysis of Results, significantly presenting all the data summarize in a precise approach the differences and similitudes of the performed research. It might represent a very useful tool for further analyses and research.
The Discussions and the Conclusions are very well structured. In this manuscript, the conclusions have been developed correctly and in a professional manner. I recommend this manuscript for publication!
Author Response
Dear reviewer
We are honored to receive your assistance and advice.
We believe that this can be successfully completed by the assistant manuscript.
Provide appropriate suggestions to help more countries or organizations facing the crisis of lack.
Wish you all the best
Reviewer 2 Report
Thank you for giving me a chance to read this instructive paper. It think it deals with an important topic and is a good fit with the journal. While I am supportive of this submission, I have five suggestions for improviding its quality.
First, my main concern relates to the presentation and content of the hypotheses: they all read exactly the same way, which I consider problematic. They should be rephrased to differ from each other. Another issue is that they state that they "assume", which is a wrong terminology for a hypothesis. A hypothesis postulates relationships but does not assume them. This is simply not correct. As a last point on the hypotheses, I suggest the authors use active and not passive voice to formulate them.
Second, I have concerns regarding the authors' discussion of water literacy, which is their main concept. The presentation of the concept appears not rigorous enough. I invite them to carefully revise this section carefully in order to provide a more comprehensive and systematic discussion of the concept.
Third, the snowball sampling used by the authors means that this is a non-probability sample. In my view, this is a serious limitation of the dataset and requires a more detailed discussion of how exactly the data was produced and why it still provides a valid basis for this study.
Fourth, table 5 is difficult to read - the authors may want to present it with a horizontal design.
Fifth, the conclusion of the paper is very self-referential. It would be useful to have a broader discussion of the implications of the findings and how they could be transferred to other countries.
Author Response
1.First, my main concern relates to the presentation and content of the hypotheses: they all read exactly the same way, which I consider problematic. They should be rephrased to differ from each other. Another issue is that they state that they "assume ", which is a wrong terminology for a hypothesis. A hypothesis postulates relationships but does not assume them. This is simply not correct. As a last point on the hypotheses, I suggest the authors use active and not passive voice to formulate them.
Dear reviewer
We appreciate your suggestions.
We have changed the narrative method of the hypothesis to strengthen the active narrative content. Such as lines 233-253.
2.Second, I have concerns regarding the authors' discussion of water literacy, which is their main concept. The presentation of the concept appears not rigorous enough. I invite them to carefully revise this section carefully in order to provide a more comprehensive and systematic discussion of the concept.
Dear reviewer
We appreciate your suggestions.
We have supplemented the discussion of water literacy and strengthened the importance of water literacy on individual issues. Such as lines 159-167; 178-180; 186-189; 192-196.
3.Third, the snowball sampling used by the authors means that this is a non-probability sample. In my view, this is a serious limitation of the dataset and requires a more detailed discussion of how exactly the data was produced and why it still provides a valid basis for this study.
Dear reviewer
We appreciate your suggestions.
We have added the process of generating snowball sampling data and the importance of this method. Such as line 330-338.
4.Fourth, table 5 is difficult to read-the authors may want to present it with a horizontal design.
Dear reviewer
We appreciate your suggestions.
We have increased the spacing of the content description in Table 5, hoping to increase the convenience of readers. As shown in Table 5.
5.Fifth, the conclusion of the paper is very self-referential. It would be useful to have a broader discussion of the implications of the findings and how they could be transferred to other countries.
Dear reviewer
We appreciate your suggestions.
We have added explanations and discussions about the contribution of research findings to other countries. Such as line 748-759.
Round 2
Reviewer 2 Report
Dear all,
I received your revised submission and could see that you did not change the wording of your three hypotheses and that you continue to use a wrong terminology.
Before recommending this paper to be published, I urge the authors to fix the following two issues:
1) Change the wording of the hypotheses in a way that they read different from each other. It is absolutely pointless of present three hypotheses that read exactly the same way.
2) It is wrong (!) to state that you "assume" a relationship if you actually want to test it. You don't test assumptions - you make assumptions.
I hope the authors will follow my recommendation this time.
Author Response
Dear reviewer
Thank you for your suggestions and reminders.
We try to understand your suggestion and do our best to correct the hypothetical statement.
Regarding the revised content of the manuscript, as shown in lines 233-251.
Hope to get your approval,
Thank you for taking the time to review
Best regards,
Round 3
Reviewer 2 Report
This manuscript is now almost ready to be accepted.
The authors only need to change table 1 to reflect the modified formulation of the hypotheses.
Author Response
Dear reviewer
Thank you for your suggestion. We have modified the narrative in Table 1 to achieve consistency with the above hypothetical description. The relevant amendments are shown in Table 1 in the manuscript.
Kind regards,
This manuscript is a resubmission of an earlier submission. The following is a list of the peer review reports and author responses from that submission.
Round 1
Reviewer 1 Report
The Introduction provides very detailed and expressive data based upon keen research and study. The innovative aspects are well highlighted, full of content, supported by relevant and up to date references.
Methods and Instruments point out in a professional manner the undertaken research, bringing a very clear view on the importance of education related to water literacy and awareness including the innovative questionnaire, which is a very good tool to extract both the issue and the solution.
The Analysis of Results, significantly presenting all the data summarize in a precise approach the differences and similitudes of the performed research. It might represent a very useful tool for further analyses and research.
The Discussions and the Conclusions are very well structured, perhaps Conclusions could be more developed.
Author Response
Reviewer 1
The Introduction provides very detailed and expressive data based upon keen research and study. The innovative aspects are well highlighted, full of content, supported by relevant and up to date references.
Thank you reviewer
Thank you for your affirmation.
We will continue to work hard and make corrections.
Methods and Instruments point out in a professional manner the undertaken research, bringing a very clear view on the importance of education related to water literacy and awareness including the innovative questionnaire, which is a very good tool to extract both the issue and the solution.
Thank you reviewer
Thank you for your affirmation.
We will continue to work hard and make corrections.
The Analysis of Results, significantly presenting all the data summarize in a precise approach the differences and similitudes of the performed research. It might represent a very useful tool for further analyses and research.
Thank you reviewer
Thank you for your affirmation.
We will continue to work hard and make corrections.
The Discussions and the Conclusions are very well structured, perhaps Conclusions could be more developed.
Thank you reviewer
Thank you for your affirmation.
We will continue to work hard and make corrections.
Thank you reviewer
Your suggestions make the manuscript more perfect. Thank you again for your assistance.
Reviewer 2 Report
In this paper, only the survey results are listed.
It is difficult to think that Taiwanese residents represent the island region.
This paper lacks a description of the response to climate change and drought.
Author Response
Reviewer 2
In this paper, only the survey results are listed.
Thank you reviewer
We have completed the corrections, such as the description of the manuscript content.
It is difficult to think that Taiwanese residents represent the island region.
Thank you reviewer
We have completed the corrections, such as the description of the manuscript content.
This paper lacks a description of the response to climate change and drought.
Thank you reviewer
We have completed the corrections, such as the description of the manuscript content.
Thank you reviewer
Your suggestions make the manuscript more perfect. Thank you again for your assistance.
Reviewer 3 Report
Dear Authors, thanks for submitting your paper to Water. While the research is a very interesting topic about Water Literacy; in my opinion, your document requires several improvements to make it publishable on Water Journal.
Please see below my comments:
(1) First of all the authors need to improve the english editing along the paper.
(2) They also need to provide better details about their methodological framework (include a one or two figures for it).
(3) Your results should also include some figures. It is hard to detect a pattern from the tables (if any). You should convert some of your large tables into figures for a better comprehension of the results.
(4) Provide a Table summarizing all tests you applied together with their assumptions and applicability.
(5) Provide a Map of Taiwan showing the distribution of questionnaries.
(6) Figure 1 must be a Tree diagram that should include all the topics included within water use awarness, water attitude, and water use behavior.
(7) The hypothesis should be improved... it is not clear in this context what does "significant difference" mean... is it a spatial difference or a temporal difference?
(8) "Study Framework and Hypothesis" section has many statements that are repeated on "Study Procedures and Instruments" ... rewrite o merge
(9) you should provide better details about the measure of scale reliability (Cronbach alpha), and why did you use it on this context?.. what does its value mean?
(10) Results: Must be improved with additional figures. Mapping the results would be ideal. Other types of figures could work as well i.e. bar plots, boxplots, etc.
(11) Discussion: The section should be named Discussion instead of Discuss. Update this section in your updated version taking into consideration the comments provided above.
(12) Conclusions. This section should also be updated in the revised version.
Author Response
Reviewer 3
(1) First of all the authors need to improve the english editing along the paper.
Thank you reviewer
We have tried our best to complete grammatical adjustments, such as the description of the manuscript content.
(2) They also need to provide better details about their methodological framework (include a one or two figures for it).
Thank you reviewer
We have made changes to improve the method structure description.
(3) Your results should also include some figures. It is hard to detect a pattern from the tables (if any). You should convert some of your large tables into figures for a better comprehension of the results.
Thank you reviewer
We have made changes to supplement the results analysis (data) description.
(4) Provide a Table summarizing all tests you applied together with their assumptions and applicability.
Thank you reviewer
We have already made supplementary explanations, corresponding to the hypothesis and test questions.
(5) Provide a Map of Taiwan showing the distribution of questionnaries.
Thank you reviewer
We have made a supplementary explanation.
(6) Figure 1 must be a Tree diagram that should include all the topics included within water use awarness, water attitude, and water use behavior.
Thank you reviewer
We have made a supplementary explanation.
(7) The hypothesis should be improved... it is not clear in this context what does "significant difference" mean... is it a spatial difference or a temporal difference?
Thank you reviewer
We have adjusted the content, such as the description of the manuscript content.
(8) "Study Framework and Hypothesis" section has many statements that are repeated on "Study Procedures and Instruments" ... rewrite o merge
Thank you reviewer
We have adjusted the content, such as the description of the manuscript content.
(9) you should provide better details about the measure of scale reliability (Cronbach alpha), and why did you use it on this context?.. what does its value mean?
Thank you reviewer
We have adjusted the content, such as the description of the manuscript content.
(2.2 Study Procedure and Instruments)
(10) Results: Must be improved with additional figures. Mapping the results would be ideal. Other types of figures could work as well i.e. bar plots, boxplots, etc.
Thank you reviewer
We have adjusted the content, such as the description of the manuscript content.
(11) Discussion: The section should be named Discussion instead of Discuss. Update this section in your updated version taking into consideration the comments provided above.
Thank you reviewer
We have adjusted the content, such as the description of the manuscript content.
(12) Conclusions. This section should also be updated in the revised version.
Thank you reviewer
We have adjusted the content, such as the description of the manuscript content.
Thank you reviewer
Your suggestions make the manuscript more perfect. Thank you again for your assistance.
Round 2
Reviewer 2 Report
All the comments have been appropriately revised.
Author Response
Dear reviewer
Your generous suggestions will make this article more complete.
Merry thank`s you
Reviewer 3 Report
The authors improved some topics revised in the first round; however, the revision made by them is only minor and it does not provide details about the requested modifications. The Authors' Responses to Reviewer's Comments are vague and do not indicate where the changes were done. I was hoping to see an improved (and different) version of this paper but I think there is still some work to do. For example: (1) Abstract: The text at the beginning does not connect to the next one. (2) Introduction. Not sure why you included additional discussion about climate change. Also, there are too many explanations of climate change that lead to study water literacy; however, there are not not supportive materials about why we should do it, or discussing why the end result of studyng water literacy is so important i.e. how it can mitigate global warming (and climate change) impacts. (3) Methods and Results: do not provide major revisions or changes compared to its intial version (see previous revision). (4) Discussion and Conclusions: Most of the text was added into these sections; however, the discussion (and also along the paper) tries to extrapolate the results of 653 questionaries to the whole Taiwan (popuation of 23.5 millions). I think you should really concentratate on how your results compare to each other i.e. gender comparisons, ages,distribution, etc. and how they relate back to water literacy advantages or disadvantages in a sicentific manner. I could reconsider my decision after a new major revision; however, the authors will have to provide a paper that is readable, easy to understanf for any audience, and with a high sicentific level on figures and discussion provided along the paper. The current version does not meet these characteristics.
Author Response
(1) Abstract: The text at the beginning does not connect to the next one.
Dear reviewer
We have made changes. Such as the abstract text.
(2) Introduction. Not sure why you included additional discussion about climate change.
Also, there are too many explanations of climate change that lead to study water literacy; however, there are not not supportive materials about why we should do it, or discussing why the end result of studyng water literacy is so important i.e. how it can mitigate global warming (and climate change) impacts.
Dear reviewer
We have added content about the importance of water literacy to energy conservation of water resources.
Please refer to 1.1, described in red font. line 114-131.
(3) Methods and Results: do not provide major revisions or changes compared to its intial version (see previous revision).
Dear reviewer
Regarding the research method, we have made the following corrections:
- We use Table 1 to explain the assumptions and problems.
- We use Figure 2 to illustrate the sample distribution of the manuscript.
- We introduce the structure and topics of the questionnaire in detail, as shown in Figure 1.
- We have revised the description of Assumption 1-Assumption 3. See the text below in Figure 1.
- We have adjusted the repetitive description content of the research framework.
- In the description of reliability measurement, we will explain all the topics. As shown in Table 1.
(4) Discussion and Conclusions: Most of the text was added into these sections; however, the discussion (and also along the paper) tries to extrapolate the results of 653 questionaries to the whole Taiwan (popuation of 23.5 millions).
Dear reviewer
We have made changes. Such as the abstract text. As shown in line 577-665.
Dear reviewer
Thank you for your suggestion. We adjust the text based on your suggestions.
Round 3
Reviewer 3 Report
The study only had about 600 questionnaries from specific regions of Taiwan. The authors should modify the statements attributing the results to the wole country. There are many statements along the paper tat need to be modified because the sample is not representative of the country.